# A Diagnostic-Driven Prospective Clinical Study Evaluating the Combination of an Antibiofilm Agent and Negative Pressure Wound Therapy

**DOI:** 10.3390/diagnostics14070774

**Published:** 2024-04-07

**Authors:** Thomas E. Serena, Emily King, Laura Serena, Kristy Breisinger, Omar Al-Jalodi, Matthew F. Myntti

**Affiliations:** 1SerenaGroup Research Foundation, Cambridge, MA 02140, USAlserena@serenagroups.com (L.S.); kbreisinger@serenagroups.com (K.B.); omarhjalodi@gmail.com (O.A.-J.); 2Next Science, LLC, Jacksonville, FL 32256, USA; mmyntti@nextscience.org

**Keywords:** pressure ulcers, biofilm, negative pressure wound therapy, pressure injuries, fluorescence imaging, wound healing

## Abstract

Background: Each year, millions of Americans develop truncal pressure ulcers (PUs) which can persist for months, years, or until the end of life. Despite the negative impact on quality of life and escalating costs associated with PUs, there is sparse evidence supporting validated and efficacious treatment options. As a result, treatment is based on opinion and extrapolation from other wound etiologies. The ideal reconstructive plan maximizes the patient’s nutritional status, incorporates the basic tenets of wound bed preparation (debridement, offloading, proper moisture balance, reduction of bacterial burden), and employs diagnostics to guide therapeutic intervention. The use of combination therapies can potentially overcome several of the barriers to wound healing. Negative pressure wound therapy (NPWT), a commonly used modality in the management of PUs, facilitates healing by stimulating the formation of granulation tissue and promoting wound contraction; however, NPWT alone is not always effective. Clinical studies examining microbial bioburden in PUs determined that most ulcers contain bacteria at levels that impede wound healing (>10^4^ CFU/g). Objective: Thus, we hypothesized that adding an anti-microbial agent to decrease both planktonic and biofilm bacteria in the wound would increase the efficacy of NPWT. Method: In this prospective study, twenty patients with recalcitrant PUs that previously failed NPWT were treated with a biofilm-disrupting agent (Blast-X, Next Science, Jacksonville, FL, USA) in combination with NPWT. Fluorescence imaging was used to follow bacterial burden and guide therapy. Results: In total, 45% of the PUs reduced in size over the course of the four-week study, with a resolution of bacterial fluorescence in the NPWT dressing and wound bed seen in an average of three weeks. Conclusion: The combination of an antibiofilm agent and NPWT reduced bacterial levels and improved wound healing in recalcitrant PUs.

## 1. Introduction

Pressure ulcers (PUs) afflict millions of Americans. The incidence rate of PUs ranges from 4 to 38% within hospital settings, with a 68% mortality rate in the elderly attributable to PUs and associated secondary complications [1]. According to a study from the US Wound Registry, less than 40% of pressure ulcers are healed at three months and the average reduction in surface area over a four-week period is less than 10% [2]. Truncal ulcers are not amendable to self-care, requiring the assistance of family members and home health nurses. In addition to the stress on patients and family members, PUs burden the health care system. In 2020, the cost of caring for PUs in the United States was approximately USD 11 billion [1].

Despite the human and financial toll inflicted by this debilitating disease, there is limited quality research on treatment [3]. Most treatment plans are founded on expert opinion due to the lack of clinical data. In addition, wound care practice has favored single therapies that are either continued or discontinued based on wound improvement. Although episodes of care in which a reconstructive plan is outlined for the entire course of therapy have been discussed, there is little supporting evidence for their implementation. The authors contend that combination therapies instituted within episodes of care that are driven by relevant diagnostics, rather than separate intervals of single therapy, may improve PU healing rates.

Most truncal PUs develop in patients with multiple comorbidities and chronic diseases [4]. Numerous systemic and local factors contribute to poor wound healing in this population [4]. Failure to address any one of these factors can result in a nonhealing wound. The ideal reconstructive plan maximizes the patient’s nutritional status, incorporates the basic tenets of wound bed preparation (debridement, offloading, proper moisture balance, reduction of bacterial burden), and employs diagnostics to guide therapeutic interventions [5].

The presence of large amounts of bacteria (>10^4^ CFU/g) inhibits wound healing [6]. In a recent clinical trial (FLAAG), nearly 91% (20/22) of PUs had bacterial loads greater than 10^4^ CFU/g [6]. In chronic wounds, bacteria favor a biofilm phenotype [7]. Biofilm bacteria attach to the wound surface, aggregate, and produce a film: the extapolymeric substance (EPS), which protects them from antiseptics and antibiotics [8]. It has been shown that disrupting the biofilm promotes healing in acute and chronic wounds [9]. The key to a biofilm-based approach to wound care includes aggressive debridement and the use of a biofilm-disrupting topical agent [10,11]. In this trial, investigators sharply debrided the wounds weekly and applied a biofilm-disrupting polyethylene glycol-based hydrogel (BlastX, Next Science, Jacksonville, FL, USA) with every negative pressure wound therapy dressing change. The gel contains a pH buffer system and a benzalkonium chloride surfactant that destabilizes the biofilm EPS and subsequently aids in killing the exposed bacteria [12]. The white gel was placed on a negative pressure sponge, as shown in Figure 1.

## 2. Materials and Methods

This study is a continuation of the parent study: “Evaluation of the combination of a biofilm-disrupting agent and negative pressure wound therapy: a case series” (9 August 2023, clinicaltrials.gov #NCT04265170) [13]. Patient demographics and results from the parent study (patients 1–10 in Table 1) were included. Doubling the sample size through the inclusion of patients from the parent study allowed for more robust analysis and refining of the primary and secondary endpoints. The quantitative secondary analysis “allows for the generation of new knowledge without the costs of administration and implementation of additional data collection and maximizes the output of large-scale studies” [14]. For detailed materials, methods, inclusion, and exclusion criteria, see the parent study in the *Journal of Wound Care* [13].

## 3. Results

Table 1 provides demographics for patients in both groups. The difference between each cohort is the withdrawal rate and trial length. Group 1 had a withdrawal rate of 40% (4/10) and group 2 had a withdrawal rate of 10% (1/10). Group 1 also completed the study on treatment day 30 while group 2 completed the study on treatment day 28. Therefore, all results for treatment day 28 are derived from group 2 and all results for treatment day 30 are derived from group 1. The study enrolled nine male (45%), ten female (50%), and one patient without gender information (5%). The age of the enrolled patients ranged from 22 years old to 82 years old, with a mean age of 65.5 years old. Group 2 had twice as many patients with less than 20 percent area reduction (PAR) of the wound compared to group 1. Group 1 had four patients with a PAR greater than 20% and two patients with a PAR less than 20%. Group 2 had five patients with a PAR greater than 20% and four patients with a PAR less than 20%. There were nine patients between both groups (45%) with PAR greater than 20%.

For the analysis, responders were classified as patients who saw a percentage area reduction greater than 20% at the end of the four-week study. Non-responders were classified as patients who did not see a percentage area reduction greater than 20% at the end of the four-week study. The overall group was classified as both responders and non-responders at the end of the four-week study. Since the length of observation was not equal for both groups (group 1 ended on treatment day 30 and group 2 ended on treatment day 28), the results for treatment days 28 and 30 were not plotted.

Primary Endpoint (wound size):

The primary endpoint is the reduction in wound surface area, with the first group evaluated through day 30 and the second group evaluated through day 28. For the ten patients enrolled in group 2, four patients (40%) had a PAR greater than 20% at four weeks. One patient in group 2 who continued treatment through day 49 had a PAR greater than 20%. As shown in Figure 2, the five patients in group 2 with a PAR greater than 20% had an average percent wound area reduction of 74%. For group 1, the four patients with a PAR greater than 20% had an average percent wound area reduction of 49%. On treatment day 28, the average percent wound area reduction for all patients in group 2 was 29%. On treatment day 30, the average percent wound area reduction for all patients in group 1 was 34%.

Figure 3 summarizes the mean surface area wound reduction for all patients in the trial. The mean surface area wound reduction for both groups combined is equal to the values reported in Figure 3.

Secondary Endpoints:

The secondary endpoint of the study was to identify changes in host proteases during therapy. Figure 4 illustrates the change in host matrix metalloprotease activity for all responders between the two groups and all patients (including those who did not respond). Patients who withdrew from the study in either group were not included. Throughout the study, a lower percentage of responders had positive MMP results than non-responders. Figure 5 is an example of the reduction in bacterial fluorescence following treatment with the biofilm disrupting agent.

## 4. Discussion

Pressure ulcers afflict roughly three million people in the United States annually [15]. Despite the gravity of the problem, there is sparse evidence supporting any given treatment regimen. There is less evidence for using a combination of therapies throughout the course of treatment.

In the United States, more than 2 million patients develop pressure ulcers annually [16], and hospital-acquired pressure ulcers account for over 60,000 deaths [17]. Despite the personal and financial cost of treating pressure ulcers, there is a paucity of evidence supporting the regimens used in treating this wound type. Treatment, largely based on experience, opinion, and extrapolation of evidence from studies on other wound types, has resulted in a healing rate of less than 30% at 3 months and a percent area reduction of approximately 10% at 4 weeks [18]. It is estimated that the annual cost to treat pressure ulcers in the US is USD 11 billion [19].

Pressure ulcers are complex wounds and there are several underlying abnormalities that lead to poor wound healing [3]. The authors hypothesize that one factor contributing to the poor healing rates is the use of a single modality rather than combination therapy. Addressing several deficits at once might promote more rapid closure. In addition, the recent advent of point-of-care diagnostics such as fluorescence imaging allows clinicians to focus treatment on specific abnormalities in the nonhealing wound. In this study, point-of-care diagnostics were used to follow combination therapy in the treatment of recalcitrant pressure ulcers.

NPWT has become a mainstay in the treatment of pressure ulcers. The evidence for NPWT does not include large randomized clinical trials; however, a meta-analysis of 16 clinical trials with a total of 629 patients demonstrated that NPWT increased healing rates and more rapid time to healing compared to the standard of care [20]. Like most therapies in wound care, NPWT does not work in every patient. Recent studies using fluorescence imaging showed that pressure ulcers contain greater than the chronic inhibitory bacterial load (CIBL) in over 80% of wounds [21]. The results from this trial demonstrated increased bacterial burden, above CIBL, in all wounds that previously failed NPWT. In addition, the NPWT sponge dressings fluoresced brightly on the first dressing changes. These findings suggested that excessive bacteria inhibited healing in these recalcitrant wounds.

Aggressive debridement and the addition of an antibiofilm agent applied at every NPWT dressing change eliminated bacterial fluorescence in all cases. Novel antibiofilm agents represent the latest advance in topical antiseptics for nonhealing wounds. In most wounds, it took 3 to 4 weeks to eliminate the fluorescence. The reduction of bacterial burden below CIBL promoted wound healing. The average PAR was 45% in wounds that had failed to heal for months.

Several studies have evaluated therapies to improve the success of NPWT [22,23]. Instillation therapy with NPWT, iNPWT (Veraflow^®^, 3M, St. Paul, MN, USA), has gained popularity in hospitals across the US [24]. In this technique, normal saline or an antiseptic solution are instilled into the wound and allowed to dwell for 10 to 20 min, after which time sub-atmospheric pressure is reapplied to the wound. The cycle is repeated three to four times per day [25]. Research suggests that iNPWT promotes wound healing and decreases bacterial burden [22]; however, iNPWT has several disadvantages: inpatient monitoring is required; additional equipment is needed to administer the installation solution; it requires greater nursing time; and it adds cost to the use of NPWT [26]. The results from this study suggest that the use of antibiofilm agents in combination with NPWT may reduce the need for this expensive therapy. Further studies are needed to confirm this supposition.

Matrix metalloprotease and human neutrophil elastase are markers of inflammation. Studies suggest that reducing bacteria in the wound will result in a concomitant fall in inflammatory markers [27]. A fall in inflammatory markers in this study corresponds to a fall in bacterial burden and more rapid healing. Further research into the relationship between host proteases and bacterial levels is warranted.

## 5. Limitations

The primary limitation of this study is the small number of patients enrolled. This trial highlights some of the challenges in conducting pressure ulcer clinical trials: most patients in the study were immobile, several patients resided in nursing homes, and many potential patients were excluded due to their inability to properly offload the ulcer and secondary to malnutrition. Several potential patients did not want to try NPWT again after failing the therapy in the past. Although patient withdrawal and drop-out are to be expected, a withdrawal rate of 20% (5/20) between the two groups resulted in a sample size that was further reduced. Statistical analysis of primary and secondary endpoints was not possible due to these factors.

## 6. Conclusions

Aggressive debridement, application of antibiofilm agents directed by point-of-care fluorescence imaging in combination with NPWT may improve healing rates in pressure ulcers that have previously failed NPWT.

## Figures and Tables

**Figure 1 diagnostics-14-00774-f001:**
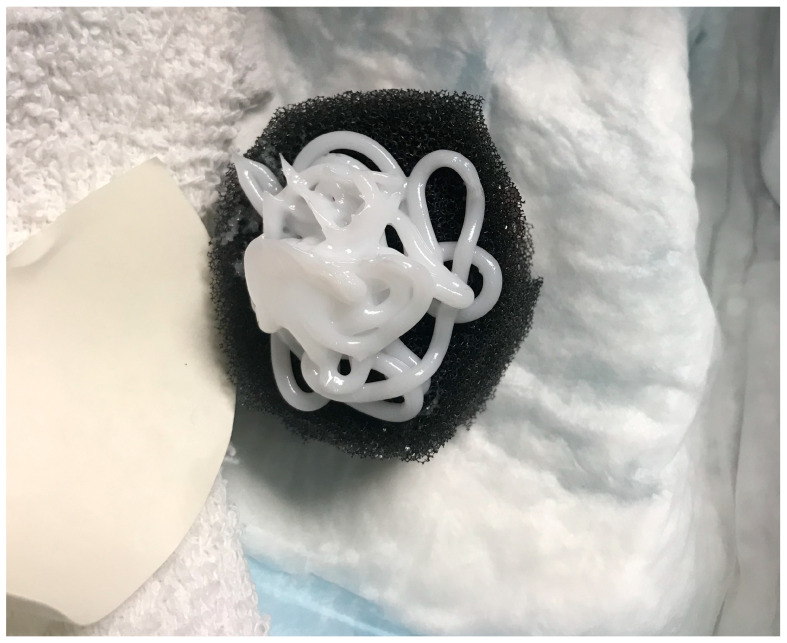
Biofilm-disrupting gel on negative pressure sponge dressing.

**Figure 2 diagnostics-14-00774-f002:**
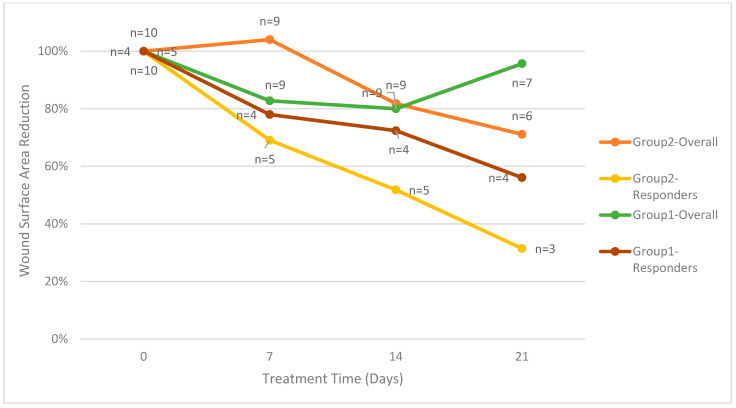
Mean wound surface area reduction divided by groups.

**Figure 3 diagnostics-14-00774-f003:**
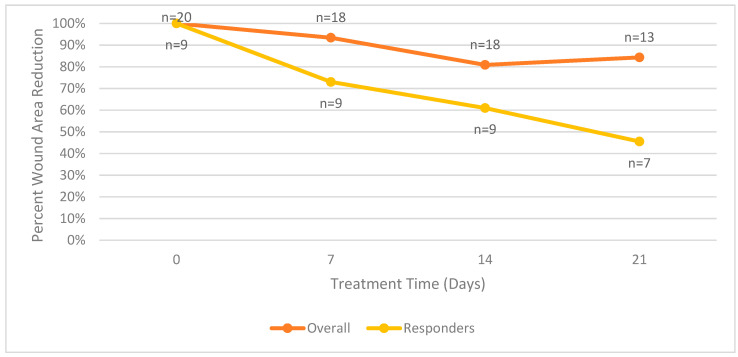
Mean wound surface area reduction comparison between overall and responder groups.

**Figure 4 diagnostics-14-00774-f004:**
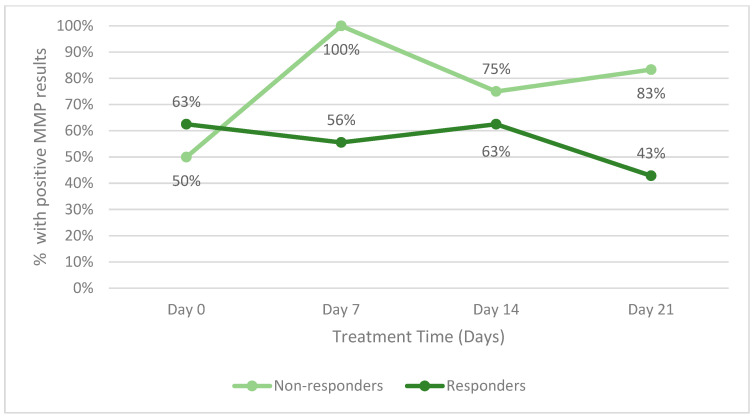
Percentage of patients with positive host matrix metalloprotease (MMP) activity within.

**Figure 5 diagnostics-14-00774-f005:**
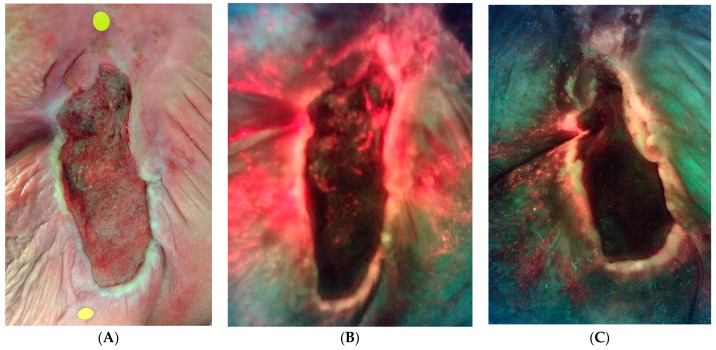
An example of bacterial fluorescence at the time of enrollment after failed negative pressure. (**A**) Standard image. (**B**) Fluorescence image with extensive bacterial fluorescence. (**C**) Reduction in ulcer surface area and marked reduction in bacterial fluorescence at 4 weeks.

**Table 1 diagnostics-14-00774-t001:** Patient demographics.

Patient	Age (Years)	Gender	Ulcer Duration (Weeks)	Mini Nutritional Assessment Score	Comorbidities (Major Conditions Listed)
1	67	Female	48	7	Type 2 diabetes, multiple traumas following a motor vehicle accident with paraplegia, permanent colostomy, hypothyroidism, anemia of chronic disease
2	70	Female	156	7	Multiple sclerosis with generalized weakness, recent bilateral hip fractures, overactive bladder
3	70	Male	68	8	Multiple sclerosis with generalized weakness, wheelchair-bound, obesity
4	46	-	58	9	-
5	59	Female	20	11	Right above-knee amputation, chronic general body pain
6	22	Male	60	12	Paraplegia, muscle spasticity, anemia of chronic disease, neuropathy
7	67	Female	52	13	Diabetes, multiple traumas following motor vehicle accident, paraplegia, history of perforated bowel with ileostomy, history of deep venous thrombosis with caval filter hypothyroidism, anemia of chronic disease
8	67	Male	8	10	Paraplegia following motor vehicle accident
9	78	Female	104	-	Diabetes, peripheral arterial disease, chronic obstructive pulmonary disease, dysphagia, encephalopathy
10	66	Female	36	8	Generalized weakness, dysphagia, coronary artery disease, hypertension, incontinence, schizoaffective disorder, lower extremity contractures, seizure disorder
11	70	Female	4	14	Type 2 diabetes, hyperlipidemia, hypothyroidism, hypertension, left breast cancer (resolved)
12	57	Female	13	14	Type 2 diabetes, hyperlipidemia, hypertension, obesity, depression
13	80	Female	7	6	Hypokalemia, chronic general body pain, constipation
14	68	Male	52	11	Hypertension, paraplegia, broken back (4 occasions), chronic general body pain, chronic migraines
15	70	Male	>52	10	Type 2 diabetes, peripheral vascular disease, lymphedema, hypertension, spinal stenosis, neuropathy, anxiety
16	82	Male	12	11	Incontinence, appendectomy, cholecystectomy, back surgery
17	69	Male	9	12	Multiple sclerosis, epilepsy, hyperlipidemia, osteoarthritis, vitamin D deficiency, vitamin B12 deficiency
18	82	Male	4	10	Type 2 diabetes, atrial fibrillation, hypertension, hypothyroidism, chronic general body pain, iron deficiency anemia
19	49	Female	208	12	Type 2 diabetes, paraplegic, hypercholesterolemia, hypertension, multiple back surgeries, urostomy ileal conduit
20	71	Male	8	7	Chronic atrial fibrillation, hypertension, benign prostatic hyperplasia, gastroesophageal reflux disease, obesity, depression, COVID-19

## Data Availability

The data from the trial are included in the manuscript and available on request.

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
