# Peer review of "A Diagnostic-Driven Prospective Clinical Study Evaluating the Combination of an Antibiofilm Agent and Negative Pressure Wound Therapy"

_diagnostics, 2024, doi:10.3390/diagnostics14070774_

Round 1

Reviewer 1 Report

Comments and Suggestions for Authors

Dear authors, 

First of all, you used an old template, from 2022. 

The idea of the study is interesting. For these patients with many comorbidities it can be the final solution to treat the PU. 

Unfortunately this article is very similar with this one: https://pubmed.ncbi.nlm.nih.gov/33439086/

At least bring 100 patients to publish something clear and relevant. 

Author Response

Thank you for reviewing this manuscript and providing feedback. The template matches the template available to download on the Diagnostics’ Instructions for Authors page. This manuscript presents a continuation of the research from the parent study “Evaluation of the combination of a biofilm-disrupting agent and negative pressure wound therapy: a case series” (clinicaltrials.gov #NCT04265170)” which is cite in this manuscript. The purpose of this manuscript is to conduct a retrospective analysis on the parent data while including the new data presented. The small sample size is discussed in Section 5. Limitations, and the challenge of enrolling patients due to comorbidities.

Reviewer 2 Report

Comments and Suggestions for Authors

The authors present a compelling comparison between the combined use of an antibiofilm agent with negative pressure therapy for treating recalcitrant pressure ulcers. Overall, the study is well-written and structured. However, there are areas where improvements could enhance clarity and depth:

1. **Abstract**: The introductory sentences in the abstract serve more as background information rather than stating the objectives directly. To address this, I recommend adding a subheading such as "Background" before delving into the objectives.

2. **Antimicrobial Agent**: It would be beneficial to specify which antimicrobial agent was used in the abstract for better clarity and understanding.

3. **Comparison between Groups**: I did not observe any comparison between different groups, such as one group receiving the gel containing the antibiofilm agent/negative pressure and another group receiving the same intervention without the antibiofilm agent/negative pressure. Incorporating such a comparison is essential for accurately interpreting the benefits of this treatment strategy.

4. **Results of Control Group**: It would be valuable to include the results of the control group (patients receiving gel without the antibiofilm agent/negative pressure) in the graphs depicting the study results. This addition would provide a comprehensive overview of the outcomes across all intervention groups.

In addressing these points, the study's findings and implications can be presented more effectively, facilitating a better understanding of the research methodology and outcomes.

Comments on the Quality of English Language

Minor editing of English language required.

Author Response

Thank you for reviewing this manuscript and providing feedback. Background and Objectives have been updated in the Abstract, as well as the name of the biofilm disrupting antimicrobial agent. Due to the small sample size, there is no control group to compare against. The results and graphs are therefore broken up as “responders” and “all patients” between the parent study and current study. These limitations and the addition of more patients would be a consideration for future studies with this biofilm disrupting antimicrobial agent.

Round 2

Reviewer 1 Report

Comments and Suggestions for Authors

The comments are relevant. 

Well done!